



# Terrestrial methane emissions from Last Glacial Maximum to preindustrial

Thomas Kleinen, Uwe Mikolajewicz, and Victor Brovkin

Max Planck Institute for Meteorology, Bundesstr. 53, 20146 Hamburg, Germany

**Correspondence:** Thomas Kleinen (thomas.kleinen@mpimet.mpg.de)

**Abstract.** We investigate the changes in terrestrial natural methane emissions between the Last Glacial Maximum (LGM) and preindustrial (PI) by performing time-slice experiments with a methane-enabled version of MPI-ESM, the Max Planck Institute for Meteorology Earth System Model. We consider all natural sources of methane except for emissions from wild animals and geological sources, i.e. emissions from wetlands, fires, and termites. Changes are dominated by changes in tropical wetland emissions, with mid-to-high latitude wetlands playing a secondary role, and all other natural sources being of minor importance. The emissions are determined by the interplay of vegetation productivity, a function of $CO_2$ and temperature, source area size, affected by sea level and ice sheet extent, and the state of the West African Monsoon, with increased emissions from north Africa during strong monsoon phases.

We show that it is possible to explain the difference in atmospheric methane between LGM and PI purely by changes in emissions. As emissions more than double between LGM and PI, changes in the atmospheric lifetime of $CH_4$, as proposed in other studies, are not required.

## 1 Introduction

The atmospheric concentration of methane undergoes major changes in the time between the last glacial maximum (LGM) and preindustrial (PI). Between LGM and 10 ka BP (before-present, with present = 1950 CE) atmospheric $CH_4$, as reconstructed from ice cores, nearly doubles from ~380 $ppb$ at LGM to 695 $ppb$ at 10 ka BP (Köhler et al., 2017), with very rapid concentration changes of about 150 $ppb$ occurring during the transitions from the Bølling Allerød (BA) into the Younger Dryas (YD) and from the YD into the Preboreal (PB) / early Holocene (Figure 1). Furthermore, while Holocene atmospheric $CH_4$ is very similar for 10 ka BP and PI (694 $ppb$, mean concentration for 300 a BP to 200 a BP), $CH_4$ decreases linearly by 15% at from 10 to 5 ka BP and increases again linearly towards PI. If we assume that the atmospheric lifetime of $CH_4$ did not change dramatically between the LGM and the present, these changes in atmospheric $CH_4$ would require large changes in $CH_4$ emissions.

The change in methane between LGM and PI has been investigated in a number of studies. Some have used box models to explain the methane changes observed in ice cores. Recently Bock et al. (2017), for example, pointed to tropical wetlands as the main driver of glacial-interglacial $CH_4$ change from a study of methane isotopes from ice cores. In addition there are studies with comprehensive models. Kaplan (2002) investigated wetland $CH_4$ emissions during the LGM and the present



using the BIOME4 model. He finds wetland emissions of $140\,Tg\,CH_4\,yr^{-1}$ $(1Tg = 10^{12}g)$ for the present-day situation and $107\,Tg\,CH_4\,yr^{-1}$ (-24%) for the LGM, with wetland areas at the LGM slightly larger than at present. Valdes et al. (2005) performed time-slice experiments with the Hadley Centre coupled model (HadCM3), using the Sheffield Dynamic Global Vegetation Model (SDGVM) as a fire and wetland methane emission model, as well as an atmospheric chemistry model. They find

PI wetland $CH_4$ emissions of $148\,Tg\,CH_4\,yr^{-1}$ and LGM emissions of $108\,Tg\,CH_4\,yr^{-1}$ (-27%), with tropical sources changing rather little and NH high latitudes contributing most of the change in emissions. Emissions from biomass burning change from $11\,Tg\,CH_4\,yr^{-1}$ at PI to $7\,Tg\,CH_4\,yr^{-1}$ at LGM (-36%), contributing to the total emission change from $199\,Tg\,CH_4\,yr^{-1}$ at PI to $152\,Tg\,CH_4\,yr^{-1}$ at LGM (-24%). Weber et al. (2010) investigated wetland emissions for PI and LGM time slices with climate forcings from the Paleo Model Intercomparison Project PMIP2 ensemble, applied to an offline wetland $CH_4$ model.

They found an overall reduction by 29-42%, with sources in the NH extratropics reduced by 51-60%, while tropical sources were reduced by 22-36%. Finally Hopcroft et al. (2017) investigated methane emission changes between LGM and PI using the Hadley Centre Global Environmental Model (HadGEM2-ES), considering wetlands, termites, biomass burning as $CH_4$ sources, along with ocean and geological emissions. They obtain an overall source reduction by 28-42%, with LGM wetland emissions ($97\,Tg\,CH_4\,yr^{-1}$, $80\,Tg\,CH_4\,yr^{-1}$ if northern peatlands considered explicitly) reduced by 30% in comparison to PI

($138\,Tg\,CH_4\,yr^{-1}$), and termite emissions reduced by 40%.

     Studies of time slices between the LGM and PI are much sparser. Kaplan et al. (2006), using BIOME4-TG as a terrestrial methane emission model also determining emissions of biogenic volatile organic compounds (BVOCs), and an atmospheric chemistry model, investigated time slices every 1000 years from LGM to the present. They found that changes in atmospheric $CH_4$ are largely due to a changed lifetime, mainly through BVOC emission changes. Interestingly, they find a larger wetland

area for the LGM than for present-day, with emissions roughly the same ($\sim110\,Tg\,CH_4\,yr^{-1}$), and an emission maximum around 10 ka BP. Finally, Singarayer et al. (2011) investigated methane for 65 time slices between 130 ka BP and PI with HadCM3 and SDGVM as a methane emission model. They point to orbital changes driving the methane increase between 5 ka BP and PI, as insolation increases in the SH tropics.

     What these studies have in common is that they require (in some cases substantial) changes in the atmospheric lifetime

of methane to explain the changes in atmospheric $CH_4$ reconstructed from ice cores. However, Levine et al. (2011) found very small changes in $CH_4$ lifetime between LGM and PI using the TOMCAT (Toulouse Off-line Model of Chemistry And Transport) atmospheric chemistry model, and Gromov et al. (2019), investigating $CH_4$ lifetime at the LGM using the EMAC model (ECHAM/MESSy Atmospheric Chemistry), also find a very similar lifetime. Therefore substantial changes in emissions are required to explain the changes in atmospheric methane.

In the present-day top-down $CH_4$ budget (Saunois et al., 2016), 59% of the emissions are from anthropogenic sources and can therefore be ignored for times before a significant human impact on the methane budget. However, 41% ($231\,Tg\,CH_4\,yr^{-1}$) of the emissions are from natural sources and are therefore relevant for the entire time since the LGM. In the top-down budget, $167\,Tg\,CH_4\,yr^{-1}$ (72% of the natural emissions) are emitted from natural wetlands, and $64\,Tg\,CH_4\,yr^{-1}$ come from "other" sources. These are not differentiated further in the top-down budget, but the bottom-up budget lists freshwater bodies (lakes),





geological sources, wild animals, wildfires, permafrost soils and vegetation as further onshore (land) sources and geological and "other" as offshore (oceanic) sources.

We aim to assess the changes in the natural sources of methane from the LGM to the present in order to determine the factors driving the changes in atmospheric CH$_4$. We use a methane-enabled version of MPI-ESM, the Max Planck Institute Earth System Model, to investigate changes in natural methane emissions for six time slices from the LGM to the present. In 65  this model we include submodels for methane fluxes from wetlands, termites and wildfires, but of the other natural methane fluxes listed above, many cannot easily be derived from the climate model state and therefore are neglected for now. As time slice experiments very likely are not helpful for looking into the BA-YD and YD-PB transitions, we neglect these for now, focusing instead on the longer timescale changes in methane.

## 2  Model and experiments

### 2.1  MPI-ESM 1.2

We use the Max Planck Institute Earth System Model (MPI-ESM) in version 1.2 (Mauritsen et al., 2019), the version to be used in CMIP6. All experiments are performed in resolution T31GR30 (Mikolajewicz et al., 2018). In comparison to the CMIP5 version (Giorgetta et al., 2013), a number of errors were corrected in the atmosphere and ocean models, and the land surface scheme JSBACH (Reick et al., 2013; Brovkin et al., 2013; Schneck et al., 2013) has been updated with a multilayer hydrology 75  scheme (Hagemann and Stacke, 2015), the SPITFIRE fire model (Thonicke et al., 2010; Lasslop et al., 2014), and the improved soil carbon model YASSO (Tuomi et al., 2009; Goll et al., 2015).

### 2.2  Wetland methane emission model

The present-day area that wetland methane emissions originate from is highly uncertain. The generation of methane in the soil is dependent on plant composition, carbon content and carbon quality, essentially ecosystem properties, as well as the degree 80  of anoxia in the soil, which depends on soil structure and water content, essentially hydrological properties. As there is no better estimate of the methane-generating area available, we determine the surface inundation and assume that this is a useful approximation of the areas where methane is generated.

### 2.2.1  Dynamic inundation model

We use an approach based on the TOPMODEL hydrological framework (Beven and Kirkby, 1979) to determine inundation 85  extent dynamically. TOPMODEL is a conceptual rainfall-runoff model, based on the compound topographic index (CTI) $\chi_i = \ln(\alpha_i / \tan(\beta_i))$ with $\alpha_i$ a dimensionless index for the area draining through point $i$ and $\beta_i$ the local slope at that point. TOPMODEL determines the local water table $z_i$ in point $i$ in relation to the grid cell mean water table $\bar{z}$:

$$z_i = \bar{z} + \frac{1}{f}(\chi_i - \bar{\chi}) \tag{1}$$





with $\chi_i$ the local CTI index in point $i$, $\bar{\chi}$ the grid cell mean CTI index, and $f$ a parameter describing the exponential decline of
transmissivity with depth. From Eq. 1 we determine the grid cell fraction with a local water table depth $z_i \geq 0$. Since inundated areas become unreasonably large in some locations, we limit the valid range of CTI values by introducing the constraint $\chi_i \geq \chi_{min}$ following Stocker et al. (2014), with $\chi_{min}$ constant in space and time. We assume this to be the inundated and therefore methane-emitting area $A_{inun}$, unless soils are frozen. In these cases we determine the fraction of liquid water in the soil $f_{liq}$ from the soil temperature $T_{soil}$

$$
f_{liq} = \begin{cases} 1 & \forall T_{soil} > 273.65\,K \\ 0.1 & \forall T_{soil} < 272.75\,K \\ (T_{soil} - 273.65\,K)\,K^{-1} & otherwise \end{cases}
$$

(limiting $f_{liq}$ to $0.1 \leq f_{liq} \leq 1$ for numerical reasons), and determine the inundated area as $A'_{inun} = A_{inun} \times f_{liq}$, reducing the inundated area under freezing conditions, as frozen soils emit less methane, similar to Gedney and Cox (2003).

To determine the grid cell mean water table position $\bar{z}$, we determine the layer saturation $\Psi_k = \Theta_k/\Theta_{fc}$ for each soil layer $k$ by dividing the volumetric moisture content $\Theta_k$ by the field capacity $\Theta_{fc}$. Starting from the bottom of the soil column, $\bar{z}$ is located in the first soil layer $l$ with layer saturation $\Psi_l$ less than the saturation threshold $\Psi_{thres}$. The final water table position then is

$$
\bar{z} = z_{b,l} - \Psi_l \Delta z_l \tag{2}
$$

with $z_{b,l}$ the bottom of soil layer $l$, and $\Delta z_l$ the height of soil layer $l$.

Values for $f$, $\chi_{min}$, and $\Psi_{thres}$ were determined from sensitivity experiments. In the experiments described here, we use $f = 2.6$, $\chi_{min} = 8.5$, and $\Psi_{thres} = 0.95$. Furthermore, comparison with remote sensing data (Prigent et al., 2012) showed that inundated area in grid cells with a mean CTI index $\bar{\chi} \leq 5.5$ is negligible. Inundation is therefore only determined for grid cells with $\bar{\chi} > 5.5$.

We use the CTI index product by Marthews et al. (2015) for the CTI index at a resolution of $15''$ in all present-day land areas, while we determine CTI index values for shelf areas that are below sea level at present, but above sea level under glacial conditions, from the ETOPO1 dataset (Amante and Eakins, 2009) using the topmodel library for R (Buytaert, 2011). In order to reduce storage requirements, we approximate the distribution of CTI values within a model grid cell by fitting a gamma distribution, following Sivapalan et al. (1987).

### 2.2.2 Wetland methane production and transport

We use the methane transport model by Riley et al. (2011) to determine wetland methane emissions, with minor modifications to adapt the model to the vegetation and carbon cycle representation in JSBACH. The Riley et al. (2011) model determines $CO_2$ and $CH_4$ production in the soil, transport of $CO_2$, $CH_4$ and $O_2$ through the three pathways diffusion, ebullition and plant aerenchyma, as well as the oxidation of methane during transport.



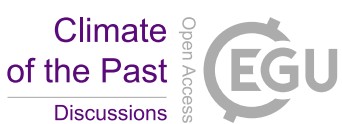

Adaptations are described in the following. In the grid cell fraction determined to be inundated by the inundation model, soil organic matter (SOM) is decomposed under anaerobic conditions in the YASSO soil carbon model (Goll et al., 2015),

assuming a reduction of decomposition by a factor of 0.35 (Wania et al., 2010) in comparison to the aerobic case. As YASSO is a zero-dimensional representation of soil C processes, we distribute the decomposition product to the soil layers according to the root distribution from Jackson et al. (1996). Partitioning of the anaerobic decomposition product into $CO_2$ and $CH_4$ is temperature-dependent, as in the original Riley model, with a baseline fraction of $CH_4$ production $fCH_4 = 0.35$ and a $Q_{10}$ factor for $fCH_4$ of $Q_{10} = 1.8$ with a reference temperature of 295K.

For each grid cell, the methane model determines $CH_4$ production and transport for two grid cell fractions, the aerobic (non-inundated) and the anaerobic (inundated) fraction of the grid cell. If the inundated fraction changes, the amounts of $CO_2$, $CH_4$ and $O_2$ are conserved, transferring gases from the shrinking fraction to the growing fraction, proportional to the area change. While vegetation in JSBACH is determined for vegetation tiles, allowing a fractional coverage of plant functional types, the relevant properties (root distribution, SOM decomposition) are aggregated to grid cell level for the methane transport model for

performance reasons. Previous sensitivity experiments showed that differences to a tile-resolving formulation are negligible.

### 2.3 Methane emissions from wildfires

To determine methane emissions from wildfires, we use the biomass burned, diagnosed from the SPITFIRE fire module (Thonicke et al., 2010; Lasslop et al., 2014), as well as information on vegetation composition from the dynamical vegetation model. We use the methane emission factors from Kaiser et al. (2012), mapped to the JSBACH plant functional types, to determine

the fraction of burned biomass emitted as methane. Therefore changes in fire-related methane emissions are completely determined by changes in fire carbon emissions. Fire occurrence in the SPITFIRE model is determined as a function of flammability (higher under dryer/warmer conditions) and ignition probability, with ignition probability a function of lightning frequency and population density. We are currently limited to a fixed lightning distribution reflecting modern conditions, and we are assuming a population density of zero for all time slices earlier than preindustrial. Therefore the main factors affecting fire-related

methane emissions are carbon content and moisture conditions.

For PI and PD we use an estimate of population density to determine the ignition probability, with ignition probability increasing with population density. However under very high population densities it is assumed that fire suppression increases, thus decreasing fire probability, and thus fire methane emissions, for very high population densities.

### 2.4 Methane emissions from termites

Methane emissions from termites are determined following the approach developed by Kirschke et al. (2013) and elaborated by Saunois et al. (2016), adapted for the use in a dynamical vegetation model. They distinguish between termite emissions from tropical and non-tropical areas, using different parameterisations for determining the termite biomass $M_{termite}$ and different emission factors for the two areas. For tropical areas, in our case defined as areas covered by the plant functional types (PFT) tropical broadleaf evergreen tree, tropical broadleaf deciduous tree, and C4 grass, we determine $M_{termite}$ from the annual gross

primary production $GPP$ using $M_{termite} = 1.21 \times \exp{(GPP \times 0.0008)}$ (Kirschke et al., 2013). From $M_{termite}$ we determine





methane emissions using an emission factor of $2.8\,\mu gCH_4/gTermite/h$ (Saunois et al., 2016). The non-tropical areas, i.e. the areas where the tropical PFTs do not occur, with termite emissions we define as the areas suitable for temperate broadleaf evergreen trees using bioclimatic limits from Sitch et al. (2003): temperature of the coldest month $T_c > 3°C$ and a growing-degree-day sum on the basis of 5°C $GDD_5 > 1200°C$. In these areas we assume a constant termite biomass $M_{termite} = 3\,g/m^2$

and an emission factor of $1.7\,\mu gCH_4/gTermite/h$ (Saunois et al., 2016). If croplands occur in any particular grid cell (not relevant for experiments presented here), emissions from the cropland tile are reduced to 40% of the non-cropland grid cell mean emissions, also following Saunois et al. (2016).

## 2.5  Model experiments

We performed model experiments for five time slices at 20 ka BP, 15 ka BP, 10 ka BP, 5 ka BP, and PI, in this case defined as

the year 1850 CE. In addition we performed one transient historical experiment for the years 1850-2010 CE, starting from the PI time slice, in order to obtain a present-day (PD) climate state for evaluation purposes. All model experiments use prescribed orbital forcing from Berger (1978) and greenhouse gas forcings from Köhler et al. (2017). Orbital parameters and greenhouse gas concentrations are supplied to the model as 10 year mean vales and are updated every 10 model years. Atmospheric aerosols were constant at 1850 conditions (Kinne et al., 2013), with the exception of the historical experiment, and we considered no

anthropogenic land use.

The time slice experiments were initialised from a – so far unpublished – transient model experiment from 26 ka BP to PI with prescribed ice sheet extent from the GLAC-1D ice sheet reconstruction (Tarasov et al., 2012; Briggs et al., 2014; Ivanovic et al., 2016). This model experiment was initiated at 26 ka BP and run transiently from then to PI, i.e. the year 1850 CE. Ice sheet extent, as well as bathymetry and topography (Meccia and Mikolajewicz, 2018) and river routing (Riddick et al., 2018)

were continuously updated throughout the deglaciation.

As the original transient experiments did not contain the methane code required for the experiments described here, the time slice experiments were initialised from the transient experiment with a three-step procedure to minimise climate drift from the original experiment. In the first step, all model components were initialised from the transient experiment, with the exception of the inundation and the methane model, which were initialised from scratch. The model was integrated for 20 years from this

initial state. This was repeated for a second time, but using the inundation and methane states reached at the end of the initial experiment. In a third step, the model was run for forty years, using the inundation and methane state reached at the end of step two, while using the conditions of the transient model experiments for all other model components. In this way we insured that the state of the physical model, as well as the biogeochemistry, would always be as close as possible to the model state in the fully transient experiment.

Present-day (PD) conditions we assess by performing a historical experiment for 1850-2010, initialised from the PI state of the transient deglaciation experiment. In the PD experiment we change GHG and atmospheric aerosol transiently, using the Stevens et al. (2017) aerosol parameterisation, but we do not consider anthropogenic land use.

Climate in the preindustrial state is very similar to the preindustrial control experiment described in Mauritsen et al. (2019) and Mikolajewicz et al. (2018). However, the orography used in the present experiments is different from that in the published




preindustrial control experiments. The latter experiments employ a mean orography, while the orography in the transient deglaciation experiment that we used as starting conditions for our time slice experiments, is an envelope orography. In the envelope orography, the grid-cell elevation is enhanced in comparison to the mean orography, in order to better represent the influence of topography on atmospheric circulation.

For all experiments we analyse a 30 year mean climatology, with the exception of the PD experiment, where we analyse a
10 year mean climatology obtained from years 2000-2009. All plots of absolute emissions are shown on the land-sea mask appropriate for the time interval under consideration, while difference plots are shown on the PI land-sea mask.

## 3 Results and discussion

### 3.1 Evaluation of present-day methane emissions

#### 3.1.1 Surface inundation

For the assessment of wetland methane emissions, the wetland area can to some extent be measured directly from satellites. Remote-sensing products of surface inundation are available, for example by Prigent et al. (2012) and Schroeder et al. (2015). To assess the quality of the modelled surface inundation, we rely on the Prigent et al. (2012) data set. However, four points need to be considered when comparing these data to model results:

1. The remote-sensing process is unable to penetrate snow cover, so snow-covered areas are considered non-inundated.

2. The remote-sensing product shows all inundated areas, including areas flooded as a result of anthropogenic processes, such as the creation of reservoirs and rice-paddies, which are not considered in the model.

3. Remote-sensing may be unable to penetrate dense forest canopy, implying that inundation estimates may be biased in forested areas.

4. Not all methane-generating areas have water tables above the surface. Water tables in northern peatlands, for example,
tend to be below the surface for part of the year, especially in the summer.

In order to make model output and remote-sensing data comparable, we therefore mask all snow-covered areas in the model output, and we use data on rice-growing areas by Monfreda et al. (2008) to mask all rice-growing areas from both the remote-sensing data and the model output. After these modifications, modelled inundated areas for the present-day period (mean over 2000-2009) are slightly larger than those observed by Prigent et al. (2012) (mean over 1993-2007) (Fig. 2). For the tropics
(TRO, here for simplicity defined as latitudes between 30°N and 30°S) the annual mean inundated area is $1.2 \times 10^6 km^2$ in the model results, while Prigent et al. show $0.8 \times 10^6 km^2$. The seasonality is phase-shifted, with the model showing the peak inundation in April, while Prigent et al. show the inundation peak in August (Fig. 2). For the glacier-free NH extratropics (NXT, here defined as north of 30°N), the seasonality of inundation is similar in observations and model, but the summer peak in inundation is larger in the model ($2.5 \times 10^6 km^2$ for the JJA mean) than in the observations ($2.3 \times 10^6 km^2$).





Comparing the spatial pattern of the annual maximum inundation (Fig. 3), the overall pattern is rather similar, although two major differences are apparent: 1) The annual maximum inundation is more localised in the observations, while it is less clearly defined and reaching lower maximum values in the model, and 2) after removal of the rice-growing areas the model does not show a significant inundation maximum in India, very likely due to an underestimate of the Indian monsoon precipitation in the low model resolution. We thus judge the methane generating areas produced by the model as reasonable, keeping in mind

the likely low bias of the remote-sensing inundation product.

    As described above, we use the inundated area to determine the methane emitting area. To evaluate the inundated areas leading to the wetland emissions, it has to be kept in mind that NXT emissions mainly are from the summer season, implying that the JJA (June, July, August) mean inundation is relevant for these, while the seasonality of TRO emissions is much less pronounced, implying that the annual mean inundation is relevant. In the following we therefore assess the effective inundated

area, defined as the annual mean inundated area in tropical latitudes (TRO, between 30°N and 30°S), and the JJA (June, July, August) mean inundated area in the glacier-free NH extratropics (NXT, north of 30°N). For the present-day climate state, the effective inundated area is $1.5 \times 10^6 \, km^2$ in TRO and $2.6 \times 10^6 \, km^2$ in NXT (differences to the numbers shown above due to the removal of rice-growing areas in the comparison to observations).

### 3.1.2   Natural methane emissions

So far it has not been possible to directly measure the quantity – surface methane fluxes – that we aim to assess in this publication on appropriate scales. Methane flux measurements exist for single sites of meter scale, mainly using measurement chambers, and for slightly larger scales, using eddy-covariance towers, but so far the scales relevant for global scale modelling, the model grid-cell to global scales, have not been covered by direct methane flux measurements (Melton et al., 2013; Saunois et al., 2016; Poulter et al., 2017). For assessment of our model experiments we therefore need to rely on global assessments

(Saunois et al., 2016), and we can gain some additional insight from atmospheric inversions (Bousquet et al., 2011).

    Under present-day (PD) climatic conditions (i.e. years 2000-2009 in the transient historical experiment), the model simulates wetland methane emissions of $222 \, Tg\,CH_4\,yr^{-1}$ ($209 - 239 \, Tg\,CH_4\,yr^{-1}$), fire emissions of $17.6 \, Tg\,CH_4\,yr^{-1}$ ($15.6 - 18.8 \, Tg\,CH_4\,yr^{-1}$), termite emissions of $11.7 \, Tg\,CH_4\,yr^{-1}$ ($10.8 - 12.2 \, Tg\,CH_4\,yr^{-1}$), and a soil uptake of $17.5 \, Tg\,CH_4\,yr^{-1}$ ($17.4 - 17.7 \, Tg\,CH_4\,yr^{-1}$). The values shown are mean values over the years 2000-2009 of the historical experiment, with

the value in brackets giving the minimum and maximum annual emissions occurring in the model results. These values fall well within the ranges reported by Saunois et al. (2016), who report $153 - 227 \, Tg\,CH_4\,yr^{-1}$ for natural wetlands, $27 - 35 \, Tg\,CH_4\,yr^{-1}$ for biomass and biofuel burning, with biofuel burning making up 30-50%, $3 - 15 \, Tg\,CH_4\,yr^{-1}$ for termites, and $9 - 47 \, Tg\,CH_4\,yr^{-1}$ for the soil uptake. Spatial patterns of modern emissions (Fig. A1 and A2) are generally similar to those shown by Saunois et al. (2016).

Furthermore, wetland methane emission estimates from atmospheric inversions (Bousquet et al., 2011) show that the majority (62-77%) of the PD emissions come from the TRO region, while a much smaller part (20-33%) are emitted from NXT. Of the modelled total wetland $CH_4$ emissions for PD conditions, $156 \, Tg\,CH_4\,yr^{-1}$ (70%) are from TRO and $65 \, Tg\,CH_4\,yr^{-1}$ (29%) are from NXT, while emissions from the SH extratropics (here defined as south of 30°S) are negligible. The latitu-





dinal distribution of modelled PD wetland methane emissions therefore is well within the range obtained from atmospheric
inversions.

Overall, the PD state is rather similar to the PI state assessed in the following section, with very small differences in the distribution of emissions (Figures A1 and A2), but generally higher methane emissions. At $287.4\,K$, the global mean temperature in the PD climate state is $0.5\,K$ warmer than preindustrial (Table 1). Precipitation is similar, leading to negligible differences in the effective inundation. With $1140\,PgC$, 8% larger than PI, the global soil C stock is also rather similar. However, vegetation
productivity is enhanced in comparison to PI, due to warmer temperatures and higher $CO_2$ concentrations. The net methane emissions in PD climate are 29% larger than PI (Table 2), with wetland methane emissions 33% larger, with a larger increase (+42%) in TRO than in NXT (+16%). Fire emissions are 18% larger than PI, termite emissions increase by 66%, and the soil uptake increases by 140%. The latter increase is largely due to the higher atmospheric concentration of $CH_4$, which drives additional methane into the soils in comparison to the lower-$CH_4$ PI state. The larger fire emissions are mainly due to higher
population densities in the 2000s than in 1850, although the very high population densities in eastern North America, Europe, and southern Asia are assumed to drive an increase in fire suppression in the SPITFIRE model (Lasslop et al., 2014). Thus fire emissions are decreased here, despite the general increase in most other places. Termite emissions are higher in the modern climate due to an increase in GPP under higher $CO_2$, while wetland emissions largely increase due to the higher temperatures of the modern climate, with $CO_2$-fertilisation playing an additional role.

## 3.2 Preindustrial methane emissions

The climate in our PI experiment is very similar to the one described by Mikolajewicz et al. (2018). The global mean near-surface air temperature is $286.9\,K$ (Table 1). The annual mean temperature in the TRO area $T_{TRO}$ is $294.5\,K$, while $T_{NXT}$, the annual mean temperature in the NXT area, is $275.2\,K$. The NH ice sheet area is limited to Greenland, with the ice sheet having an area of $1.8 \times 10^6\,km^2$ in our model setup. Under these climatic boundary conditions, we obtain total net methane
emissions of $181\,TgCH_4\,yr^{-1}$ (Table 2), with wetlands contributing $167\,TgCH_4\,yr^{-1}$ (Fig. 4a), fire and termites $15\,TgCH_4\,yr^{-1}$ and $7.0\,TgCH_4\,yr^{-1}$, respectively (Fig. 5a and b), while the soil uptake is $7.3\,TgCH_4\,yr^{-1}$ (Fig. 5c).

Wetland emissions, the dominant natural component of the terrestrial methane fluxes, mainly originate in TRO ($110\,TgCH_4\,yr^{-1}$), while emissions from NXT are $56\,TgCH_4\,yr^{-1}$ (Table 2). The main factors determining wetland methane fluxes, apart from temperature, are the emitting area and the soil carbon stock that the soil respiration (and thus methane production) is derived
from. In the PI state the global effective inundated area is $4.0 \times 10^6\,km^2$ (Fig. 4b), of which $2.7 \times 10^6\,km^2$ are located in NXT, while $1.3 \times 10^6\,km^2$ are in TRO (Table 1). For soil carbon, on the other hand, the global stock is $1054\,PgC$ ($1Pg = 10^{15}g$), with most of the soil carbon ($588\,PgC$) located in NXT, while it is $439\,PgC$ in TRO.

Methane emissions from fires (Fig. 5a) closely follow the fire distribution, with most fire methane emissions coming from subtropical Africa and South America, although some emissions also originate in North America, Southern Europe and South
Asia. Termite emissions, on the other hand mainly originate from tropical regions, especially southern Asia, with minor contributions from subtropical regions on all continents (Fig. 5b). Methane uptake by upland soils (Fig. 5c), finally, is distributed widely with no large regional variations.





### 3.3 Wetland methane emissions

Under LGM boundary conditions the global mean temperature is $4.4\,K$ colder than under PI conditions (Table 1). Extensive
glaciers cover the NH extratropics and sea level is lower, leading to a 15% increase in total land area, although total glacier-free
area is nearly identical (Table 1). The TRO area $A_{TRO}$ thus is 12% larger than PI, while the NXT area $A_{NXT}$ (by definition
glacier-free) is 14% smaller. The temperature decrease is less pronounced in TRO ($-3.1\,K$) than in NXT ($-5.8\,K$). Precipitation
decreases by 10% in the global mean, with an 11% decrease in TRO and a 19% decrease in NXT. TRO effective inundated
area $I_{TRO}$ thus increases by 19% (Table 1, Figure 6 a, Figure A3 a), while NXT effective inundated area $I_{NXT}$ decreases by 6%.
The global soil C stock is $617\,PgC$, substantially smaller (-41%) than at PI, with the decrease smaller in TRO (-33%) than in
NXT (-49%). As a result of these climate changes, wetland methane emissions decrease by 51% (Table 2, Figure 6 e, Figure
A3 e), with a TRO emission decrease of 47%, while NXT emissions decrease by 59%, with the majority of the latter emissions
coming from areas in East Asia adjacent to the Yellow Sea and North America south of the Laurentide ice sheet. The wetland
$CH_4$ emissions therefore decrease nearly everywhere (Figure 6 e), with one major exception: The continental shelf areas that
are exposed due to the lower sea level become significant sources of methane, especially in Indonesia, but also in Africa and
Asia. At $29\,TgCH_4\,yr^{-1}$, they contribute about 35% of the total wetland $CH_4$ emissions at the LGM.

For 15 ka BP, the global mean temperature change is $-2.8\,K$, relative to PI. NH ice sheet extent is 24% lower than at LGM,
but still extensive. The total land area is 12% larger than PI due to the lower sea level, but $A_{nonglac}$ in only larger by 1%.
$A_{NXT}$ is thus reduced by 10% (Table 1), while $A_{TRO}$ is increased by 11%. The change in $T_{TRO}$ is $-2.0\,K$ (Table 1), while it
is $-3.8\,K$ for $T_{NXT}$. Precipitation decreases by 6% in the global mean, with a a 4% decrease in TRO and a 10% decrease in
NXT. Precipitation in NH Africa is slightly increased due to a stronger West African monsoon. $I_{TRO}$ thus is larger by 29%,
while $I_{NXT}$ is 8% smaller than PI (Table 1, Figure 6 b, Figure A3 b). Global soil C is at 815 PgC (-23%), with TRO C stocks at
398 PgC (-9%), while NXT stocks are at 392 PgC (-33%). Total wetland methane emissions decrease by -22% as a result, with
TRO emissions decreasing by 17% and NXT emissions of by 33% (Table 2, Figure 6 f, Figure A3 f). In contrast to the LGM
situation, there is an increase in $CH_4$ emissions from NH (sub-) tropical Africa Figure 6 f) to $19\,TgCH_4\,yr^{-1}$ (+58%), due to
wetter conditions in the Sahel area. The exposed shelf areas emit about $38\,TgCH_4\,yr^{-1}$ overall, 29% of the total emissions.

For 10 ka BP, our model indicates a global mean temperature change of $-0.7\,K$ (Table 1). Glacial area is much reduced in
comparison to the LGM, but remains of the Laurentide ice sheet still cover parts of north-eastern Canada, leading to a lower
sea level than at PI. The total land area thus is 3% larger than at PI ($A_{nonglac}$ -1%), with $A_{TRO}$ 3% larger due to lower sea level
and $A_{NXT}$ 4% smaller due to the remaining ice sheet coverage. The temperature decrease is larger in TRO ($-1.0\,K$) than in
NXT ($-0.6\,K$). Precipitation is near PI levels in the global mean (-1%), with a 6% increase in TRO, and a 1% decrease in NXT.
Precipitation in NH Africa is strongly increased due to a strong West African monsoon. $I_{TRO}$ is increased by 35%, mainly
in due to the wetter conditions in north Africa, while $I_{NXT}$ is decreased by 12% in NXT (Table 1, Figure 6 c, Figure A3 c).
Global soil C is at 983 PgC, quite near the PI total stock (-7%), with a TRO soil C stock of 449 PgC (+2%) and a NXT stock
of 510 PgC (-13%). As a result, wetland $CH_4$ emissions are very similar to PI (Table 2), with +7% in TRO wetland emissions
and -14% in NXT emissions (Table 2, Figure 6 g, Figure A3 g). The increase in TRO emissions mainly occurs in the Sahel





area, where the West African monsoon is strongly increased, leading to more precipitation, increased inundated area, and more biomass and soil C. Emissions from NH Africa are $37\,Tg\,CH_4\,yr^{-1}$ (+208%), an increase larger than the total increase in TRO emissions. Emissions from the (small) exposed shelf areas are at $8\,Tg\,CH_4\,yr^{-1}$ (5% of the total wetland $CH_4$ emissions). NXT

emissions are smaller than PI in North America and Europe, but they are larger than PI in northern Asia, due to the summer warming from the changed insolation at 10 ka BP.

At 5 ka BP, global mean temperature change is at $-0.2\,K$ (Table 1). Ice sheet areas are as in the PI state, thus TRO and NXT areas are unchanged. $T_{TRO}$ is slightly lower ($-0.6\,K$), but $T_{NXT}$ is very similar to PI. Precipitation changes are very small in the global mean, with a a 4% increase in TRO and a 2% increase in NXT, with the West African monsoon slightly stronger than

PI. $I_{TRO}$ increases by 15%, while $I_{NXT}$ decreases by 1% (Table 1, Figure 6 d, Figure A3 d). Global soil C stocks are 1043 PgC, slightly smaller than preindustrial (-1%), with an increase by 4% in TRO, especially in the southern Sahel region, while NXT is 4% lower than PI. Total wetland methane emissions increase by 2%, with TRO and NXT wetland emissions both increasing by 2% (Table 2). Emissions from NH Africa are $21\,Tg\,CH_4\,yr^{-1}$ (+75% compared to PI, 19% of TRO emissions), while emissions from the SH are generally decreased. NXT emissions are decreased in northern North America, while emissions from northern

Asia and southern North America are increased.

## 3.4 Methane emissions from wildfires

For all time slices before PI we assume that no humans were present, leading to a generally decreased probability of fire ignition in comparison to PI and PD. For the LGM (Fig. 7 a) fire $CH_4$ emissions (Table 2) are 73% smaller. As biomass is reduced strongly under the cold and low-$CO_2$ conditions of the LGM, fire-related C emissions are also reduced. At 15 ka

BP (Fig. 7 b) fire emissions are 53% lower than PI, while they are 40% smaller at 10 ka BP (Fig. 7 c). Generally, the spatial pattern of emission changes at 15 and 10 ka BP mainly reflects precipitation changes: Enhanced emissions occur in areas where precipitation is reduced, enhancing vegetation flammability. In the Sahel area, this relationship is different, though. Here, the enhanced rainfall leads to an increase in vegetation cover, especially grass cover. As a result, more biomass is available for combustion, leading to enhanced emissions. At 5 ka BP, finally, fire emissions are reduced by 45%. As climate is already

relatively similar to the PI situation, the main reason for the fire emission reduction here is the smaller ignition probability due to the absence of humans.

## 3.5 Methane emissions from termites

Termite emissions are mainly determined by gross primary productivity (GPP) in tropical and subtropical areas. The lower atmospheric $CO_2$ and temperature under LGM conditions decrease GPP everywhere. Therefore termite $CH_4$ emissions are

reduced by 58% relative to the PI level (Table 2). For 15 ka BP, there also is a general reduction in termite emissions, with $4.6\,Tg\,CH_4\,yr^{-1}$ in total (-34%). However, the enhanced rainfall in the Sahel area leads to an increase in termite methane in this area. The latter is similar at 10 ka BP, where total emissions are 13% smaller in comparison to PI. The enhanced productivity in the Sahel therefore more than compensates the decrease in termite methane from the Amazon and African rain forests. At





5 ka BP, finally, termite emissions are slightly smaller than PI (-7%), with minor decreases in the rain forest areas and a slight
increase in the Sahel.

### 3.6   Methane uptake by soils

The soil continually exchanges methane and oxygen with the atmosphere through diffusion. In areas where soil conditions are
aerobic, methane concentrations in the soil are smaller than atmospheric concentrations, thus driving a flux of methane into
the soil. In the soil the methane is oxidised, with oxidation rates dependent on the concentrations of $CH_4$ and $O_2$, as well as
temperature. The gas exchange between soil and atmosphere is also modified by the presence of plants, as some plant tissues
can transport gases between plant roots and leaves.

In our experiments, we find that the soil uptake of methane is to a large extent determined by the gradient of methane between
soil and atmosphere. Thus higher atmospheric concentrations of methane directly lead to a larger soil uptake of methane. Under
LGM conditions, the atmospheric $CH_4$ concentration is $370\,ppb$, slightly less than half the PI concentration. Consequently soil
methane uptake decreases by 68% compared to PI (Table 2), with decreased temperatures being an additional factor (Fig. 9 a).
At 15 ka BP (atmospheric $CH_4$ of $464\,ppb$), the soil uptake is 52% smaller, while it is changed by -14% at 10 ka BP ($688\,ppb$)
and -28% at 5 ka BP ($579\,ppb$). Spatially, the change in methane uptake is rather uniform, showing a similar reduction in
uptake in most locations (Fig. 9). The exception to this is, once again, the Sahel area, which shows an increase in methane
uptake most pronounced for 10 ka BP (Fig. 9 c), but also for 5 ka BP (Fig. 9 d). For these time slices the increase in vegetation
cover in the Sahel region leads to a localised increase in methane uptake.

### 3.7   Time slice comparison

The net natural methane flux, i.e. the sum of all flux components, increases from $86\,Tg\,CH_4\,yr^{-1}$ at 20 ka BP (-52% compared
to PI) to $181\,Tg\,CH_4\,yr^{-1}$ in the PI state, and $233\,Tg\,CH_4\,yr^{-1}$ (+29%) at present.

The wetland emissions from TRO are the most important component of the net methane flux during all time slices. Their
contribution is smallest at PI (61% of total net emissions) and largest at 20 ka BP (67%). The contribution from NXT ranges
from 27% at 20 ka BP to 32% at 5 ka BP. Fire emissions make up 4-5% in the purely natural states between 20 ka BP and 5 ka
BP, and about 8% for the anthropogenically influenced states at PI and PD. Termite emissions make up between 3.4 and 5.0%
of net emissions, and soil uptake reduces the emissions by between 2.5% at 15 ka BP and 7.0% in the PD state (Table 2).

In the modelled emissions, we are missing two components of the natural methane cycle: Wild animals and geolog-
ical sources. For geological emissions, estimates vary widely, with bottom-up estimates in Saunois et al. (2016) of $35 -$
$76\,Tg\,CH_4\,yr^{-1}$ for on- and offshore sources, while Petrenko et al. (2017), estimating methane [14]C for the YD from ice cores,
constrain methane stemming from old carbon reservoirs to the range $0 - 18.1\,Tg\,CH_4\,yr^{-1}$ at the YD. These fluxes likely are
constant in time, although there might be changes during periods of sea level rise and fall, as hypothesised for $CO_2$ by Huy-
bers and Langmuir (2009). Methane emissions from wild animals, especially ruminants, are very difficult to estimate, current
estimates for the present span a range $2 - 15\,Tg\,CH_4\,yr^{-1}$ (Saunois et al., 2016), and estimates for other time slices are even
less confident, but might be of the order of $15 - 20\,Tg\,CH_4\,yr^{-1}$ for times before significant human influence (Chappellaz et al.,





1993). In principle, these emissions should somehow be related to the net primary productivity, as this would determine the carrying capacity of the ecosystem, implying smaller fluxes in the glacial than in the Holocene. Adopting the ice-core based estimate by Petrenko et al. (2017) for the geological fluxes, we can thus hypothesise these to be $9 \pm 9\,Tg\,CH_4\,yr^{-1}$, while wild animals might add $15 \pm 10\,Tg\,CH_4\,yr^{-1}$. The total unaccounted fluxes might therefore be of the order of $24 \pm 19\,Tg\,CH_4\,yr^{-1}$.

To compare the net fluxes to the reconstructed atmospheric $CH_4$ concentrations from ice cores, we determined the implied methane emissions (Fig. 10). We converted the methane concentrations into a methane burden, using a conversion factor of $2.767\,Tg\,CH_4\,ppb^{-1}$ (Dlugokencky et al., 1998). With a tropospheric lifetime of $9.3\,yrs$ (range given $7.1 - 10.6\,yrs$), an approximation for the present-day situation (Saunois et al., 2016), we then determined the methane flux required to obtain the $CH_4$ concentration reconstructed for all time slices except the present-day period, which is dominated by anthropogenic emissions. From this flux we subtracted the unaccounted sources, as described above, to determine the implied emissions. Uncertainties from the unaccounted fluxes are represented as error bars, however the uncertainty in tropospheric lifetime is not considered here, but would roughly add another 15%.

Comparing the modelled net emissions to the implied emissions (Fig. 10), the modelled fluxes are within the range of uncertainty for all time slices except for 15 ka BP and 5 ka BP, with modelled net emissions larger than the implied fluxes for these time slices. The net emissions increase by more than 100% going from 20 ka BP to 10 ka BP and PI, we can thus explain the methane increase from LGM to Holocene with $CH_4$ emissions only, not requiring changes in methane lifetime. However, we so far cannot explain the Holocene changes in atmospheric $CH_4$, decreasing between 10 ka BP and 5 ka BP, and increasing subsequently. We assume that this is due to an overestimate of the West African monsoon and its impact on African methane emissions, as a general reduction in the West African monsoon would lead to decreases in TRO emissions for 15 ka BP, 10 ka BP, and 5 ka BP, bringing model results more in line with the implied emissions determined from ice core $CH_4$. However, this is speculative at this point and would require further experiments.

## 4 Conclusions

In this assessment we considered all natural emissions of methane, with the exception of emissions from wild animals and geological sources. In our experiments we found that it is possible to explain the difference between LGM (20 ka BP) and PI methane concentrations purely by changes in the emissions of methane, without requiring changes in the atmospheric lifetime of $CH_4$. The time slice experiments we performed suggest that there are three main drivers to changes in methane emissions over the time from the LGM to the present:

1. Global mean temperature and $CO_2$: Higher atmospheric $CO_2$ concentrations increase NPP and thus also soil carbon available for anaerobic decomposition to $CH_4$. Similarly, higher global mean temperature also increases NPP and soil C decomposition, and it furthermore increases the ratio of $CH_4$ to $CO_2$ production in anaerobic decomposition. Thus, higher atmospheric $CO_2$ and higher global mean temperature lead to larger wetland emissions of $CH_4$. This affects emissions from wildfires and termites in a similar way, as fire C release is dependent on biomass and termite biomass is dependent on GPP and thus $CO_2$ and temperature.

2. Ice sheet area and sea level: Larger ice sheets remove $CH_4$ sources in the northern hemisphere extratropics as these are covered by the ice sheets, which is especially important for wetland methane emissions from North America. At the same time large ice sheets lower sea level, enlarging tropical wetland area as the continental shelf is exposed and becomes a significant source of methane. This is mainly relevant in the tropics, as high latitude shelf areas exposed under glacial conditions, for example the Laptev sea shelf, experience extremely cold conditions in glacial climate,

leading to negligible methane emissions. Exposed shelf areas in Indonesia, Africa, and South America, on the other hand, emit significant amounts of methane. Thus lower sea level leads to larger emitting areas and thus higher emissions of methane.

3. The West African monsoon: During the time slices when the West African monsoon is stronger than at present, i.e. at 15 ka BP, 10 ka BP, and 5 ka BP, precipitation in the Sahel region is significantly enhanced in comparison to the PI

state, leading to an increase in vegetation cover, productivity and biomass burning. As a result, methane emissions at these times are stronger than at present, leading to a significant increase in (sub-)tropical $CH_4$ emissions, with all natural methane sources increased.

For methane emissions from wildfires, a further factor influencing the emissions is the human population density, as this strongly affects the fire probability in the SPITFIRE model employed in JSBACH (Fig. 10). The soil uptake of methane, on the

other hand, is strongly dependent on the atmospheric concentration of methane (Fig. 10).

The changes in methane from LGM to the present are dominated by changes in tropical wetland emissions, with mid and high latitude wetland emissions being a significant but secondary factor, gaining in importance as the high latitudes become ice-free. In total the wetland emissions account for $93 - 96\%$ of the net $CH_4$ flux, and all other methane sources are of minor importance.

*Code and data availability.* The primary data, that is the model code for MPI-ESM, is freely available to the scientific community and can be accessed with a license on the MPI-M model distribution website. In addition, secondary data and scripts that may be useful in reproducing the authors' work are archived by the Max Planck Institute for Meteorology. They can be obtained by contacting the first author or publications@mpimet.mpg.de.

**Appendix A: Overview over absolute emissions**

*Author contributions.* TK developed the methane module, performed model experiments and wrote the manuscript. UM performed the transient model experiment the climate states were derived from and contributed to climate model development. VB contributed to methane module development. TK and VB planned the study, and all authors discussed the analysis and the manuscript.





*Competing interests.* The authors declare that they have no conflict of interests.

*Acknowledgements.* We acknowledge support through the project PalMod, funded by the German Federal Ministry of Education and Re-
search (BMBF), Grant No. 01LP1507B, under the FONA research for sustainability initiative (www.fona.de). Computational resources were
made available by Deutsches Klimarechenzentrum (DKRZ). We thank Anne Dallmeyer for comments on an earlier version of the manuscript.



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





**Tables and Figures**

| Time | $T_{GM}$ | $T_{TRO}$ | $T_{NXT}$ | $A_{nonglac}$ | $A_{TRO}$ | $A_{NXT}$ | $I_{TRO}$ | $I_{NXT}$ |
|---|---|---|---|---|---|---|---|---|
| 20 ka BP | 282.5 | 291.5 | 269.4 | 133.9 | 72.3 | 54.3 | 1.5 | 2.5 |
| 15 ka BP | 284.1 | 292.5 | 271.4 | 135.2 | 71.7 | 56.6 | 1.6 | 2.5 |
| 10 ka BP | 286.2 | 293.6 | 274.6 | 133.2 | 66.1 | 60.6 | 1.7 | 2.4 |
| 5 ka BP | 286.7 | 294.0 | 275.2 | 134.0 | 64.5 | 62.7 | 1.5 | 2.7 |
| PI | 286.9 | 294.6 | 275.2 | 134.1 | 64.5 | 63.0 | 1.3 | 2.7 |
| present | 287.4 | 295.1 | 276.0 | 134.1 | 64.5 | 63.0 | 1.5 | 2.6 |

**Table 1.** Climate and areas in experiments. Global mean annual temperature $T_{GM}$, TRO temperature $T_{TRO}$, and NXT temperature $T_{TRO}$, all in $K$. Global non-glaciated land area $A_{nonglac}$, TRO area $A_{TRO}$, NXT area $A_{NXT}$, TRO effective inundated area $I_{TRO}$, and NXT effective inundated area $I_{NXT}$, all in $10^6 km^2$.

| Time | soil sink | wetland | fire | termite | net | TRO | NXT |
|---|---|---|---|---|---|---|---|
| 20 ka BP | -2.3 | 81.8 | 4.0 | 3.0 | 86.4 | 58.1 | 23.3 |
| 15 ka BP | -3.5 | 129.6 | 7.0 | 4.7 | 137.7 | 91.5 | 37.7 |
| 10 ka BP | -6.2 | 165.8 | 8.8 | 6.1 | 174.5 | 117.3 | 48.3 |
| 5 ka BP | -5.2 | 170.7 | 8.1 | 6.5 | 180.1 | 112.6 | 57.7 |
| PI | -7.3 | 166.7 | 14.9 | 7.1 | 181.4 | 110.1 | 56.3 |
| present | -17.6 | 221.5 | 17.6 | 11.7 | 233.4 | 155.9 | 65.4 |

**Table 2.** Methane emissions for all time slices in $Tg CH_4 yr^{-1}$. Shown are soil uptake, total wetland emissions, fire and termite emissions, net emissions, and wetland emissions from TRO and NXT.





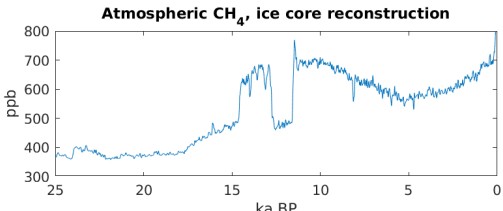

**Figure 1.** Atmospheric CH₄ as reconstructed from ice cores (Köhler et al., 2017).

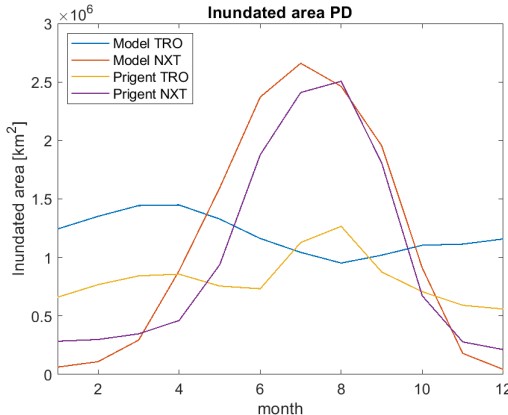

**Figure 2.** Climatology of monthly mean inundated area for model years 2000-2009 and Prigent et al. observations 1993-2007, separated for Tropics (TRO) and NH extratropics (NXT).





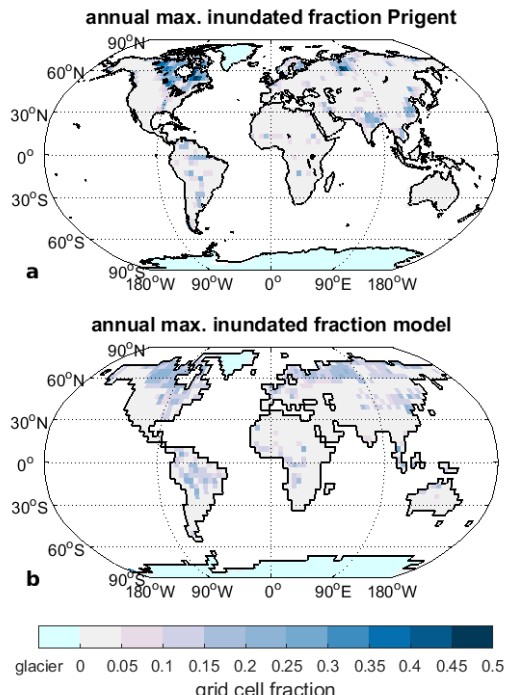

**Figure 3.** Annual maximum of mean monthly inundated fraction for Prigent et al. (a) and model (b).

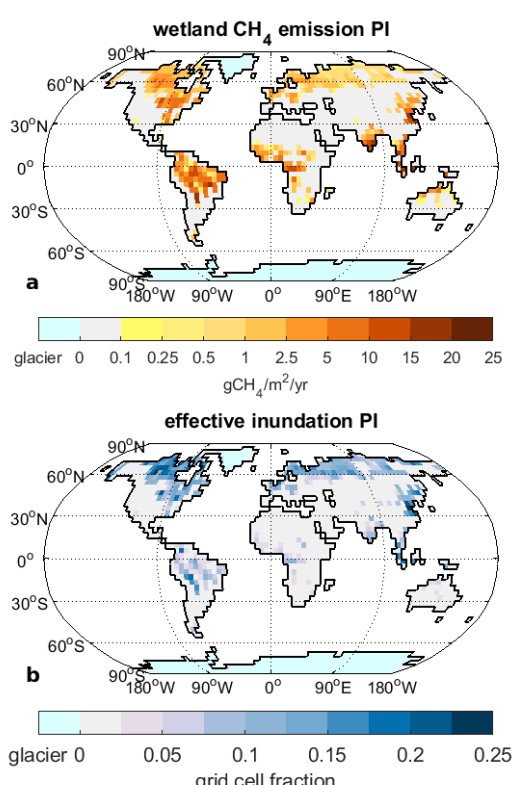

**Figure 4.** Wetland $CH_4$ emissions for preindustrial (PI) climate: Annual emissions of $CH_4$ from wetlands (a) and effective inundation (b).



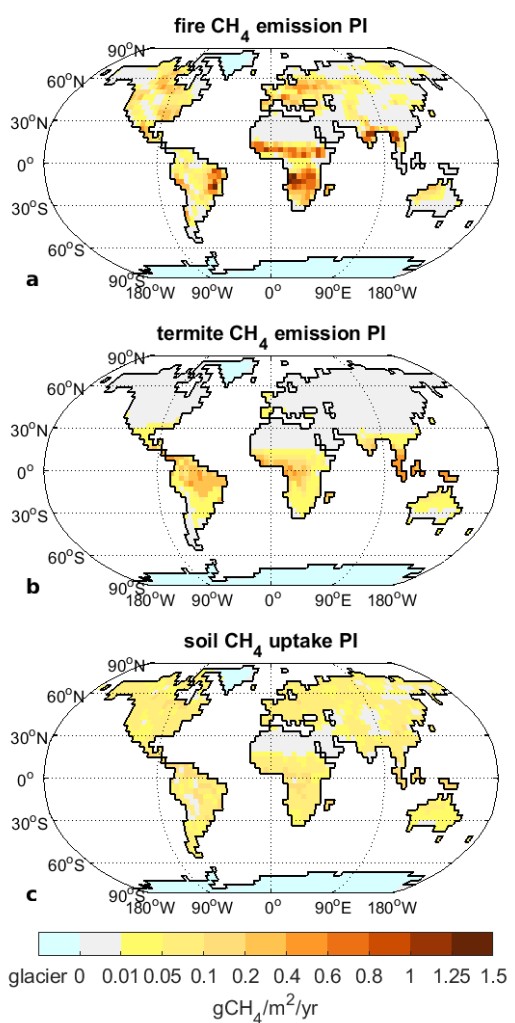

**Figure 5.** Non-wetland CH$_4$ emissions for PI climate: Annual emissions of CH$_4$ from fires (a) and termites (b), as well as annual soil uptake of CH$_4$ (c).

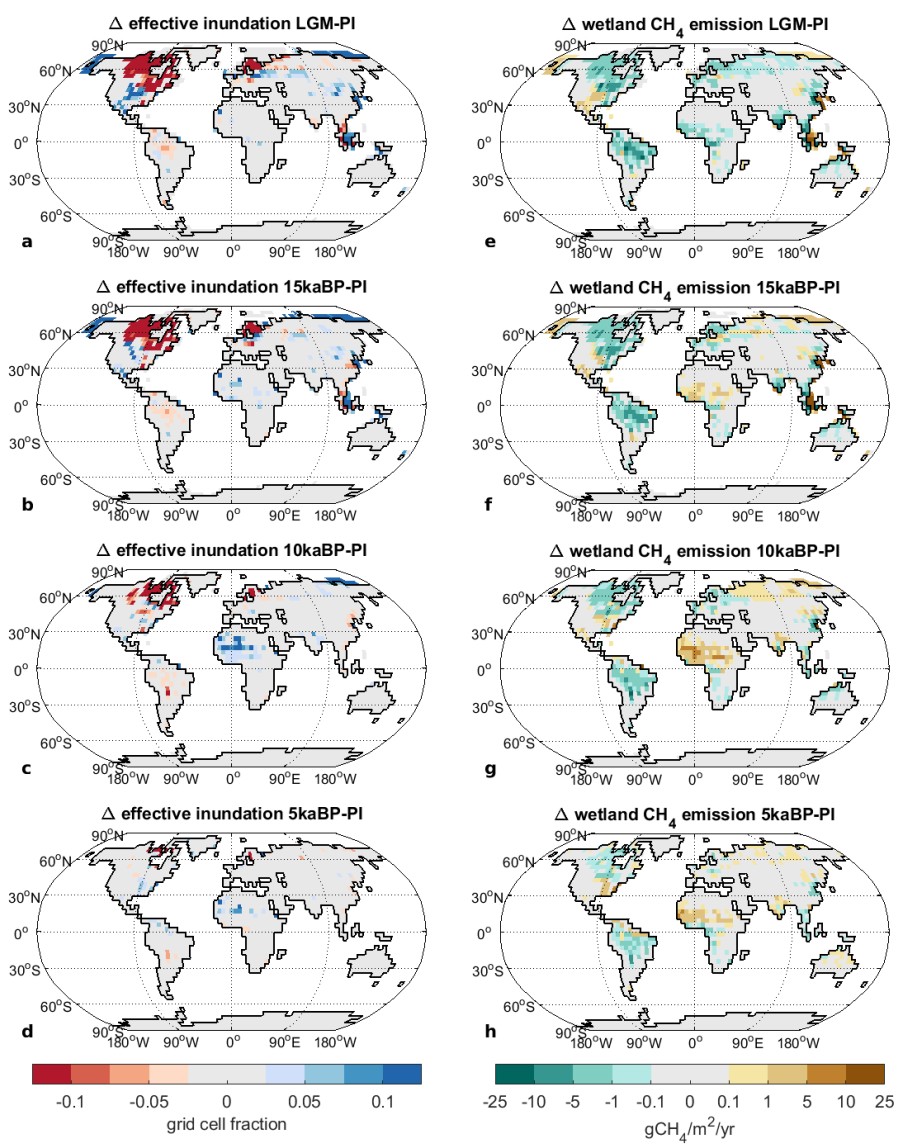

**Figure 6.** Change in effective inundation and wetland methane emissions for past climate states: a-d inundation difference to PI, e-h CH$_4$ emission difference to PI. a,e: LGM; b,f: 15 ka BP; c,g: 10 ka BP; d,h: 5 ka BP.

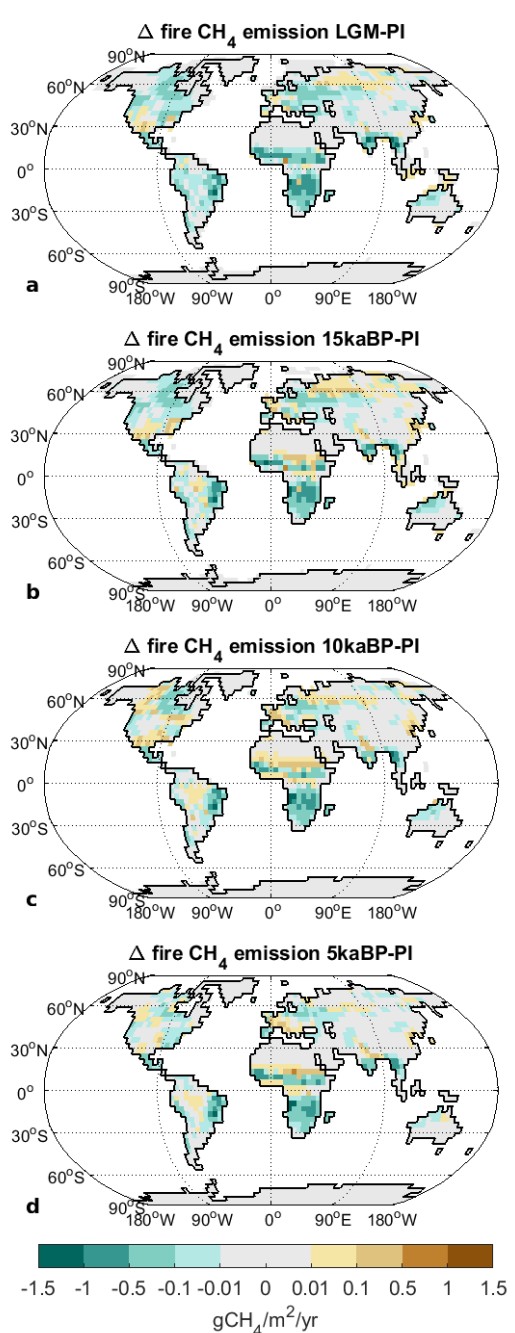

**Figure 7.** Difference in wildfire methane emissions to preindustrial for a) LGM, b) 15 ka BP, c) 10 ka BP, and d) 5 ka BP.



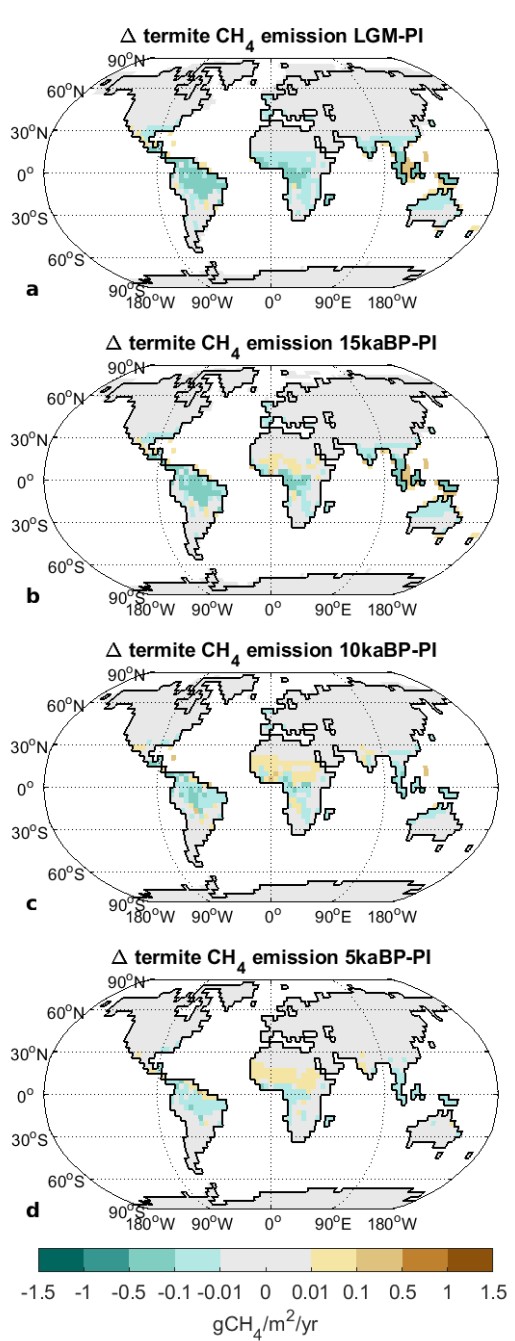

**Figure 8.** Difference in termite methane emissions to preindustrial for a) LGM, b) 15 ka BP, c) 10 ka BP, and d) 5 ka BP.



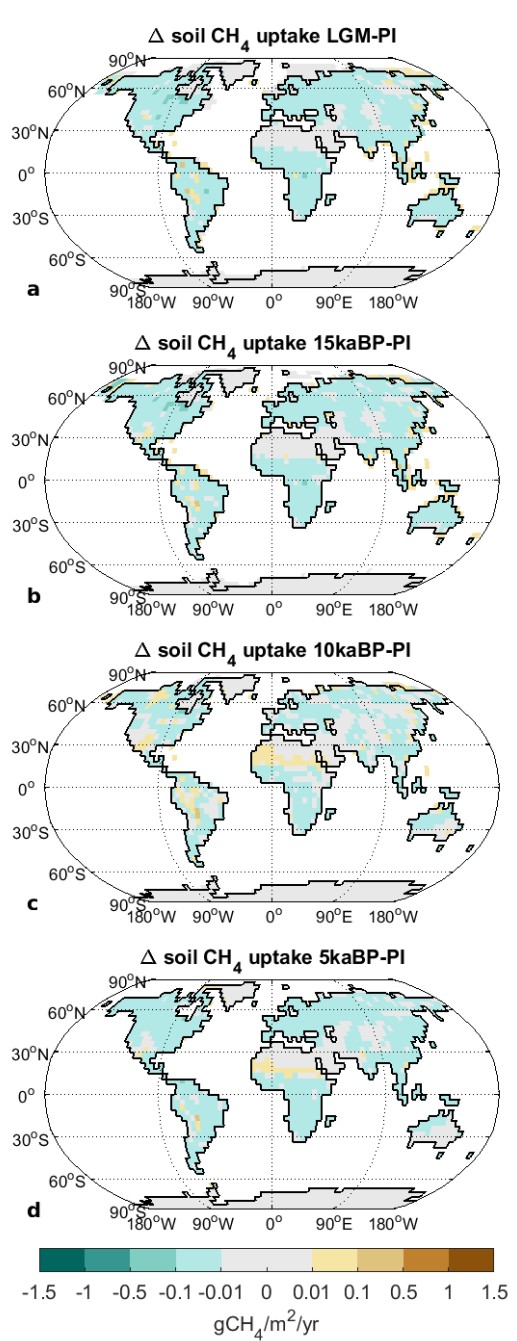

**Figure 9.** Difference in methane soil uptake to preindustrial for a) LGM, b) 15 ka BP, c) 10 ka BP, and d) 5 ka BP.





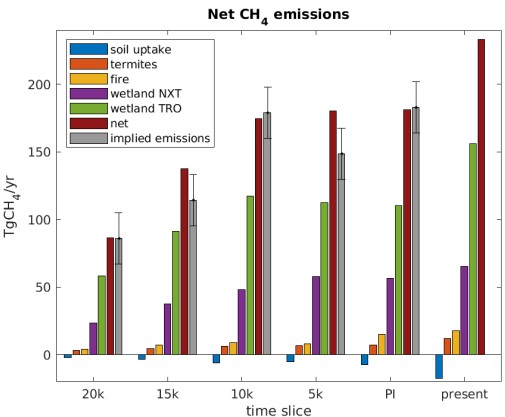

**Figure 10.** Components of the net CH$_4$ emissions for all timeslices. Soil uptake of CH$_4$ is shown as a negative flux. Calculation of implied emissions and error bar as detailed in the text.

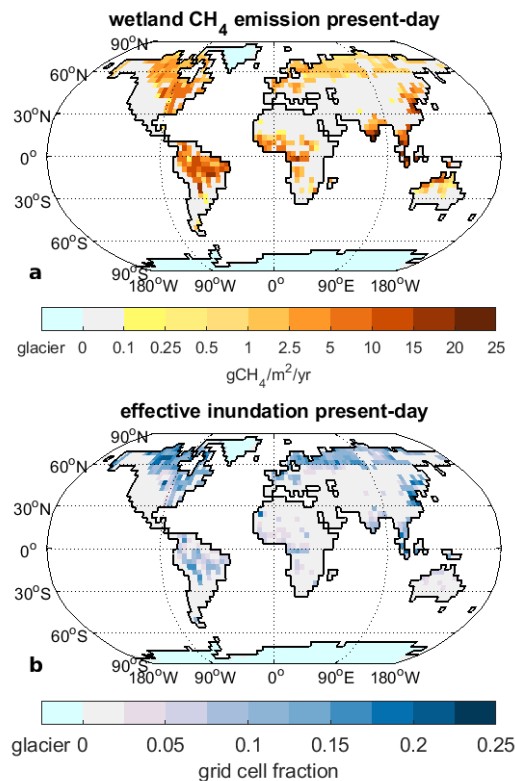

**Figure A1.** Wetland CH$_4$ emissions for present-day climate (2000-2009): Annual emissions of CH$_4$ from wetlands (a) and effective inundation (b). Please note the different colour scale for (b) in comparison to Fig. 3.

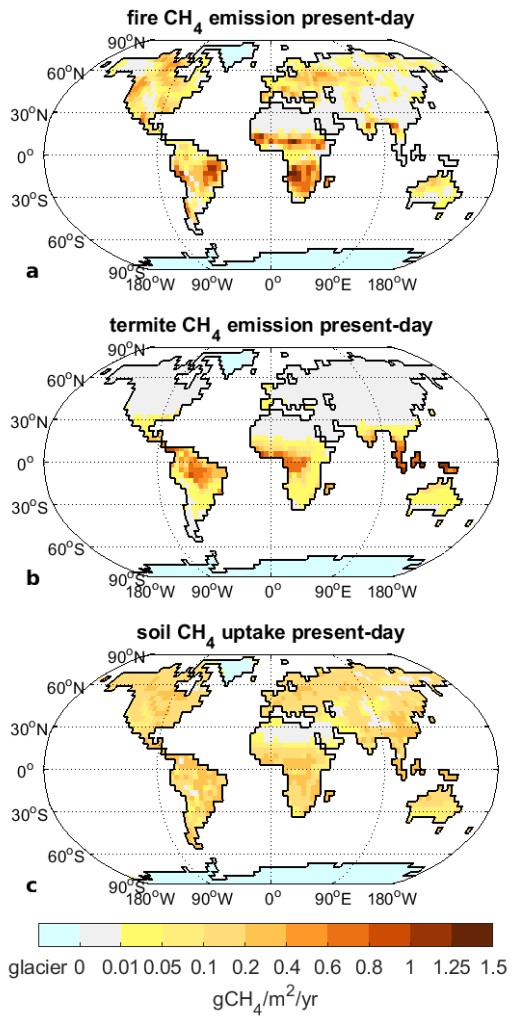

**Figure A2.** Model results for present-day climate (2000-2009): Annual emissions of CH$_4$ from fires (a) and termites (b), as well as annual soil uptake of CH$_4$ (c).



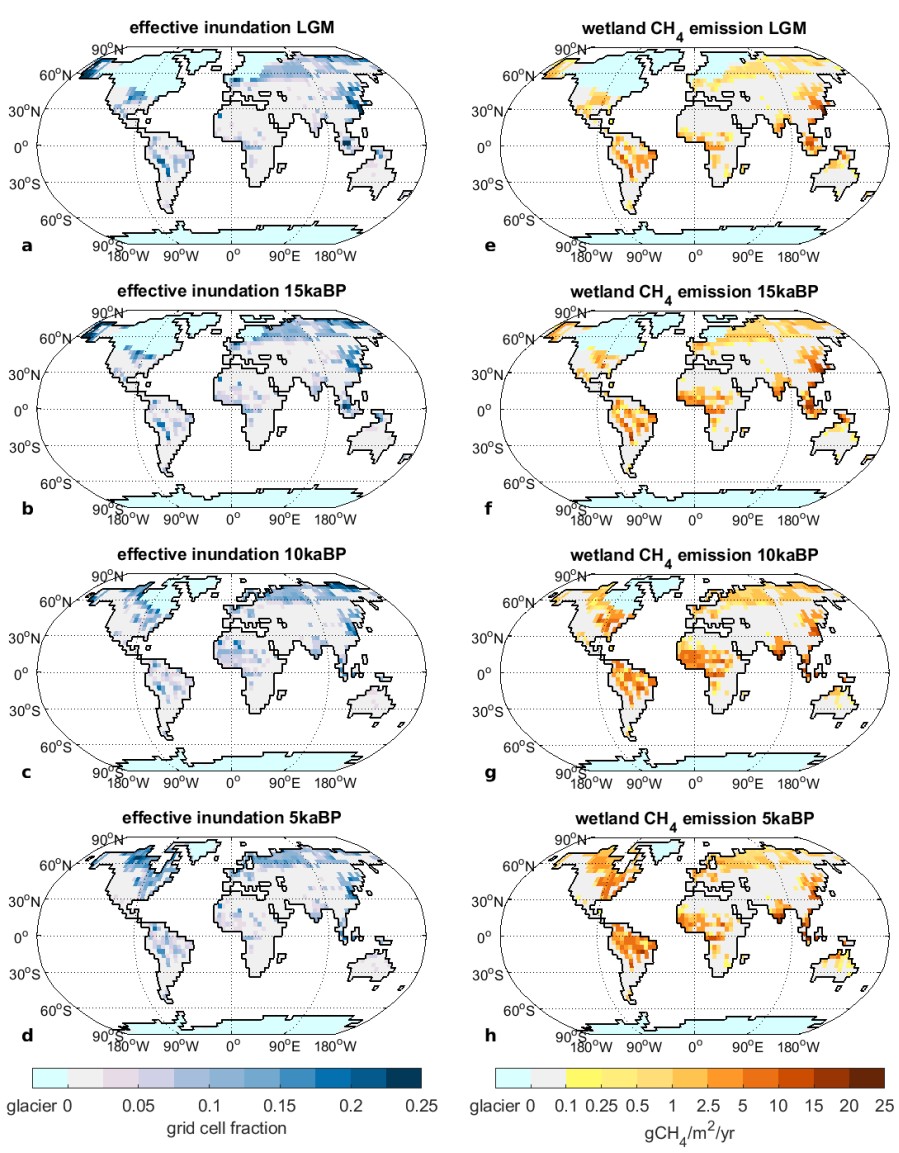

**Figure A3.** Absolute effective inundation and wetland CH$_4$ emissions for past climate states: a-d effective inundation , e-h wetland CH$_4$ emission. a,e: LGM; b,f: 15 ka BP; c,g: 10 ka BP; d,h: 5 ka BP.