# Peer review of "Terrestrial methane emissions from Last Glacial Maximum to preindustrial"

_Climate of the Past, 2019_

## Referee Comment (RC1) · Anonymous Referee #1 · 8 Oct 2019

**Summary**

The authors present a comprehensive suite of ESM simulations including methane emissions. They focus in 5 ka intervals from the LGM to the present day. They show that emissions are reduced by around 50% at the LGM relative to the PI, meaning that no atmospheric lifetime change is required to explain the observed low concentration at the LGM, consistent with 3 recent studies of the atmospheric chemistry of the LGM. This is the first study to reconcile the emissions required by the observed CH4 drop without requiring to a large change in lifetime. Hence, these results will be important for understanding CH4 during ice-ages. I recommend minor revisions as explained below.

**Main comments**

[Figure]

To me, the main results of the paper require more explanation. If the model is able to correctly resolve a 50\% emissions reduction at the LGM relative to the PI, there may be different reasons for this, e.g.: (i) the simulated LGM climate is less favourable for emissions than in older studies; (ii) the CH4 model is more sensitive to LGM conditions than in older studies, either through a different tuning, or because of increased model realism/complexity. (iii) the coupling of the methane model to the ESM increases the sensitivity to LGM conditions. In my opinion these options need to be clearly evaluated, otherwise we only know that it is possible to get this 50% reduction but not why or how.

Moreover, these results don't show increasing southern hemisphere emissions over the past 6ka that would explain the observed recovery of atmospheric CH4, thus meaning that a human influence is not required. It would be good to understand how and why your results don't replicate this result from Singarayer et al., 2011. You discuss this a bit in lines 397-402, but I believe this could benefit from more detail, without needing further simulations.

Technical comments

Line 1: Sorry to be picky, but past tense would make more sense: "Underwent", "nearly doubled" etc.

Line 50: "However, Levine et al. (2011) found very small changes in CH4 lifetime between LGM and PI"

You could mention here that this has also been found by Murray et al 2014, and Hopcroft et al 2017, using different atmosphere-chemistry models.

It would also be worth pointing out here why the lifetime didn't change. i.e. because of balance of competing influences from changes in reactive compounds like isoprene and reduced reaction rates due to lower temperatures, and lower generation of OH from lower water vapour levels.

Line 54: Here you can set up the main finding of you paper, by pointing out here that

none of these earlier studies have managed to get the right change in emissions and lifetime together.

Line 177: insured -> ensured

Line 313: delete 'in'

Line 397: This is really interesting, but why/how does your model manage this and previous studies did not? I realise this is complex, but are there any obvious difference in your simulated LGM state? Is it drier in critical areas? Or is it due to the added complexity of your wetland module?

Line 399: This is an interesting finding. What is the explanation though? Do you see the same reduction in precipitation and hence wetland flux in Southern hemisphere as found by Singarayer et al 2011?

Line 433: I think I understand the meaning of this sentence: "In total the wetland emissions account for $93-96\%$ of the net CH4 flux, and all other methane sources are of minor importance." , but perhaps you could reword it.

Figures: It is slightly surprising to see Termite CH4 emissions in Europe.

References L. Murray, L.J. Mickley, J.O. Kaplan, E.D. Sofen, M. Pfeiffer and B. Alexander (2014). Factors controlling variability in the oxidative capacity of the troposphere since the Last Glacial Maximum, Atmos. Chem. Phys. 14, 3589-3622.

---

## Referee Comment (RC2) · Anonymous Referee #2 · 20 Oct 2019

Manuscript summary: Thomas Kleinen et al. present an analysis of changes in methane fluxes from wetlands, termites and wildfires since the LGM. The analysis is performed using the Max Planck Institute for Meteorology Earth System Model, which explicitly simulates methane emissions (and the soil sink). Time-slice experiments are performed in the model, at 5 kyr intervals beginning at 20 kyr. The model is also run for the present day and compared with best available methane budget assessments. The authors find that wetland methane emissions dominated the changes in atmospheric methane over this time, and that tropical wetlands were the most important component of this.

Overall assessment and major comments:

It is difficult for me to assess the technical aspects of the MPI-ESM work, as I do not

work with ESMs myself; I hope that another reviewer is able to do this. That said, the provided descriptions suggest a comprehensive and well-grounded approach, and the MPI Meteorology group does very good work in my opinion. The model simulates present-day methane emissions that are reasonable and generally compare well with top-down and bottom-up constraints. The model also produces methane emissions that appear to be mostly consistent with the ice core atmospheric methane record.

My main concern with this submission to CP is its relative lack of novelty. I view CP as one of the leading journals publishing on paleoclimate, and as such I think that successful submissions to this journal should add substantially to our understanding of some aspect of paleoclimate. The major finding of the paper (that tropical wetland emissions were the main factor driving the LGM - PI atmospheric methane change) has been argued for many times previously, including by model-based studies. While there have been studies arguing for other factors (e.g., the Kaplan et al 2006 study the authors cited), the leading role of tropical wetlands is the most accepted explanation. I think additional model results are valuable, even if they only reinforce the currently accepted hypothesis, but I'm not sure that CP is the best place – Earth System Science Data may be a better fit for this kind of study.

It may be possible that the work described in this manuscript is much more technically advanced than prior efforts. In this case, a publication in CP may be warranted, but the authors should then make a very clear argument for why their model is superior to what has been done before, and is expected to produce the most reliable results.

Additional comments: I would recommend the addition of ice core constraints regarding the methane interpolar gradient (e.g., Baumgartner et al., 2012, Biogeosciences) into the analysis – is the partitioning between tropical and extratropical sources in the model consistent with these constraints?

Page 7, last paragraph (around line 210). The disagreement between model results and satellite observations for surface inundation is discouraging. I would recommend

more discussion regarding how much uncertainty / error this could potentially introduce into the model wetland emissions estimates.

Minor comments: Line 15 – 17. The Oldest Dryas – Bolling was an interval of similarly rapid methane change, I recommend mentioning this

Paragraph around line 50. I would recommend adding the GESO-Chem LGM and PI results of Murray et al., 2014, ACP into the discussion of methane lifetime.

---

## Author Response (AR1)

**Dr. Thomas Kleinen**

Abteilung Land im Erdsystem

Max-Planck-Institut für Meteorologie

Bundesstr. 53

20146 Hamburg

Deutschland

Tel.: +49 - (0)40 - 41173 - 140

Fax: +49 - (0)40 - 41173 - 350

thomas.kleinen@mpimet.mpg.de

www.mpimet.mpg.de

Max-Planck-Institut für Meteorologie | Bundesstr. 53 | 20146 Hamburg

Ed Brooks

c/o Climate of the Past

**Revision of manuscript cp-2019-109**

Dear Ed,

Hamburg, den 9. Januar 2020

unfortunately the revision of our manuscript has taken substantially longer than we had anticipated. Contrary to our expectations, we could not use the offline land surface model to do the sensitivity experiments, but had to redo all sensitivity experiments using the full Earth System Model.

I apologize for the delay – and am very glad I am finally able to send the revised manuscript to you.

As you will be able to see, the main change to the manuscript is that we performed additional sensitivity experiments to better explain the reasons for the change in methane emissions between LGM and PI. We analyse the results in an additional appendix (now appendix A, the old one became appendix B) and added some text to the experiment description section, as well as to the section on the wetland $CH_4$ emissions.

Of course there are further smaller changes in response to the reviewers' comments, which I will detail below.

To this letter I have appended a detailed response to the reviewers' comments and a manuscript version that shows all changes. Please do let me know if you believe further changes to the manuscript will be necessary, or if you need any additional clarification.

Thank you very much.

All the best,
Thomas Kleinen

PS: I will contact you separately by email about the James Lee et al. manuscript – I didn't think it necessary for this manuscript, but we are working on another one where we will also discuss the interpolar gradient in more detail, and there I will definitely want more information.

**Reply to Reviewers**

We very much thank the reviewers for taking the time to review our manuscript. I have included the reviewer's comments in bold font, while our our original reply is in normal font. Finally, what we have changed in the manuscript in order to address the reviewer's point is explained in blue colour.

**Reviewer #1**

**The authors present a comprehensive suite of ESM simulations including methane emissions. They focus in 5 ka intervals from the LGM to the present day. They show that emissions are reduced by around 50% at the LGM relative to the PI, meaning that no atmospheric lifetime change is required to explain the observed low concentration at the LGM, consistent with 3 recent studies of the atmospheric chemistry of the LGM. This is the first study to reconcile the emissions required by the observed CH4 drop without requiring to a large change in lifetime. Hence, these results will be important for understanding CH4 during ice-ages. I recommend minor revisions as explained below.**

**Main comments**

**To me, the main results of the paper require more explanation. If the model is able to correctly resolve a 50% emissions reduction at the LGM relative to the PI, there may be different reasons for this, e.g.: (i) the simulated LGM climate is less favourable for emissions than in older studies; (ii) the CH4 model is more sensitive to LGM conditions than in older studies, either through a different tuning, or because of increased model realism/complexity. (iii) the coupling of the methane model to the ESM increases the sensitivity to LGM conditions. In my opinion these options need to be clearly evaluated, otherwise we only know that it is possible to get this 50% reduction but not why or how.**

Yes, we agree that this is the major shortcoming of our manuscript. While we will not be able to explain why older studies did not get similar results – we do not have access to their models after all – we should be able to shed some more light on the reasons why our model does what it does.
This will require further model experiments however, in order to isolate the different factors. At the time we originally wrote the manuscript, we did not have a working offline configuration of the land surface model available, so we couldn't perform the necessary sensitivity experiments. In the mean time, we have been able to solve the issues with the offline model, allowing us to run the necessary sensitivity experiments.
We have addressed this point by reviewer #1 by performing additional sensitivity sensitivity experiments. As it turned out that soils in the offline model are substantially dryer then in the online model, we could not use the offline land surface scheme, as originally thought, but had to perofrm additional experiments with the full ESM. Results of the experiments are analysed in appendix A, and additional text explaining the experiments and reporting the main results was added to sections 2.5 and 3.3.

**Moreover, these results don't show increasing southern hemisphere emissions over the past 6ka that would explain the observed recovery of atmospheric CH4, thus meaning that a human influence is not required. It would be good to understand how and why your results don't replicate this result from Singarayer et al., 2011. You discuss this a bit in lines 397-402, but I believe this could benefit from more detail, without needing further simulations.**

We will clarify this in the revised version.

Not having access to Singrayer's original results make a detailed comparison impossible. However we have added some discussion of this in section 3.7. We also see a decrease in SH emissions for 5 ka BP, but the change in total emissions for the time slice is dominated by the increase in North African emissions due to the monsoon increase, which likely is too strong in our model, although we are glad to finally get a stronger monsoon (including a green Sahara), something most other models do not reproduce.

**Technical comments**

**Line 1: Sorry to be picky, but past tense would make more sense: "Underwent", "nearly doubled" etc.**

Thanks for being picky – we will address this.

Changed in text.

**Line 50: "However, Levine et al. (2011) found very small changes in CH4 lifetime between LGM and PI"**
**You could mention here that this has also been found by Murray et al 2014, and Hopcroft et al 2017, using different atmosphere-chemistry models. It would also be worth pointing out here why the lifetime didn't change. i.e. because of balance of competing influences from changes in reactive compounds like isoprene and reduced reaction rates due to lower temperatures, and lower generation of OH from lower water vapour levels.**

Thanks, we will include that in the revision.

Added to text.

**Line 54: Here you can set up the main finding of you paper, by pointing out here that none of these earlier studies have managed to get the right change in emissions and lifetime together.**

Very good point by the reviewer, we did exactly that.

**Line 177: insured -> ensured**

Changed in text.

**Line 313: delete 'in'**

Changed in text.

**Line 397: This is really interesting, but why/how does your model manage this and previous studies did not? I realise this is complex, but are there any obvious difference in your simulated LGM state? Is it drier in critical areas? Or is it due to the added complexity of your wetland module?**

As detailed above, we will perform some additional offline sensitivity experiments to address this. My assumption is that it's a combination of the relatively large change in soil carbon stocks and the temperature sensitivity of methane production that we included, which is relatively new and likely wasn't included in at least some of the previous modeling studies.

Addressed by additional sensitivity experiments.

**Line 399: This is an interesting finding. What is the explanation though? Do you see the same reduction in precipitation and hence wetland flux in Southern hemisphere as found by Singarayer et al 2011?**

We will analyse this in more detail and discuss it in the revised paper.
See above – briefly discussed in section 3.7.

**Line 433: I think I understand the meaning of this sentence: "In total the wetland emissions account for 93−96% of the net CH4 flux, and all other methane sources are of minor importance." , but perhaps you could reword it.**

Thank you for pointing this out, we will clarify it.
We have reformulated the sentence – I hope it is clearer now.

**Figures: It is slightly surprising to see Termite CH4 emissions in Europe.**

We agree. However it is what we get when we implement the termite model from the Kirschke et al. / Saunois et al. reviews. We may add a short discussion of this to the section.
In the end we did not add any text on this, as Saunois et al. also show emissions in Europe in their paper – our model reproduces their distribution, so we didn't think any additional comment was required.

**References L. Murray, L.J. Mickley, J.O. Kaplan, E.D. Sofen, M. Pfeiffer and B. Alexan- der (2014). Factors controlling variability in the oxidative capacity of the troposphere since the Last Glacial Maximum, Atmos. Chem. Phys. 14, 3589-3622.**

**Reviewer #2**

**Manuscript summary: Thomas Kleinen et al. present an analysis of changes in methane fluxes from wetlands, termites and wildfires since the LGM. The analysis is performed using the Max Planck Institute for Meteorology Earth System Model, which explicitly simulates methane emissions (and the soil sink). Time-slice experiments are performed in the model, at 5 kyr intervals beginning at 20 kyr. The model is also run for the present day and compared with best available methane budget assessments. The authors find that wetland methane emissions dominated the changes in atmospheric methane over this time, and that tropical wetlands were the most important component of this.**

**Overall assessment and major comments:**
**It is difficult for me to assess the technical aspects of the MPI-ESM work, as I do not work with ESMs myself; I hope that another reviewer is able to do this. That said, the provided descriptions suggest a comprehensive and well-grounded approach, and the MPI Meteorology group does very good work in my opinion. The model simulates present-day methane emissions that are reasonable and generally compare well with top-down and bottom-up constraints. The model also produces methane emissions that appear to be mostly consistent with the ice core atmospheric methane record. My main concern with this submission to CP is its relative lack of novelty. I view CP as one of the leading journals publishing on paleoclimate, and as such I think that successful submissions to this journal should add substantially to our understanding of some aspect of paleoclimate. The major finding of the paper (that tropical wetland emissions were the main**

**factor driving the LGM - PI atmospheric methane change) has been argued for many times previously, including by model-based studies. While there have been studies arguing for other factors (e.g., the Kaplan et al 2006 study the authors cited), the leading role of tropical wetlands is the most accepted explanation. I think additional model results are valuable, even if they only reinforce the currently accepted hypothesis, but I'm not sure that CP is the best place – Earth System Science Data may be a better fit for this kind of study.**

**It may be possible that the work described in this manuscript is much more technically advanced than prior efforts. In this case, a publication in CP may be warranted, but the authors should then make a very clear argument for why their model is superior to what has been done before, and is expected to produce the most reliable results. Additional comments: I would recommend the addition of ice core constraints regarding the methane interpolar gradient (e.g., Baumgartner et al., 2012, Biogeosciences) into the analysis – is the partitioning between tropical and extratropical sources in the model consistent with these constraints?**

We very much thank the reviewer for the overall praise that we read from her or his comments. However we have to disagree in some aspects:

Yes, the reviewer is perfectly correct that our finding that tropical wetlands are the dominant source of methane is not novel in itself. However, to our knowledge nobody has been able to show this in Earth System Model results, certainly not in a setup as internally consistent as ours.

We are able to show that we get reasonable emissions for the present-day situation, including a latitudinal distribution that is consistent with atmospheric inversions. Most other studies that we are aware of were not able to show this. We are also able to show that our emissions for other time slices are reasonable, in the sense that they are similar enough to ice core reconstructions to fall within a quantified uncertainty range, and we do not require major adjustments of the atmospheric lifetime of methane in order to achieve this.

We therefore argue that our results are more technically advanced than previous efforts. We will also add a more thorough analysis on the reasons why we do get these better results than previous studies, as detailed in the reply to referee #1.

The reviewer's point about the interpolar gradient, however, we regard as a very good suggestion, we will certainly take it up in the revision.

We addressed the point about the novelty and relevance of our results by adding a sentence in the introduction section, "none of the previous studies have managed to obtain the required changes in methane emissions, while fulfilling the constraints of the present-day methane budget at the same time."

We have also added a discussion of the implications of the interpolar gradient in section 3.7, citing Baumgartner and Mitchell.

**Page 7, last paragraph (around line 210). The disagreement between model results and satellite observations for surface inundation is discouraging. I would recommend more discussion regarding how much uncertainty / error this could potentially introduce into the model wetland emissions estimates.**

We will attempt to do so. Unfortunately the current state of remote sensing of inundation under closed canopies, as in tropical rainforests, leaves much to be desired, and major discrepancies exist between different remote sensing products. We acknowledge that we did discuss this well, we will attempt to improve it for the revised version.

We have attempted to improve this, but are somewhat unhappy with the result – we think we clarified this a little, but not comprehensively, but doing better would have required a lengthy discussion of

remote sensing products, their uncertainties and their interpretation, and we felt the issue does not warrant a lengthy discussion as it isn't the focus of the present paper.

**Minor comments: Line 15 – 17. The Oldest Dryas – Bolling was an interval of similarly rapid methane change, I recommend mentioning this**

Thank you for reminding us, we will do so.
Added to the introduction section.

**Paragraph around line 50. I would recommend adding the GESO-Chem LGM and PI results of Murray et al., 2014, ACP into the discussion of methane lifetime.**

Yes, thank you for reminding us. Somehow the Murray et al. reference was lost in one of the previous revisions of the manuscript, we will add it again.
Added to the introduction section.

[revised manuscript text omitted]